# Association of CYP26C1 Promoter Hypomethylation with Small Vessel Occlusion in Korean Subjects

**DOI:** 10.3390/genes12101622

**Published:** 2021-10-14

**Authors:** Eun-Ji Lee, Myung-Sunny Kim, Nam-Hui Yim, Min Ho Cha

**Affiliations:** 1Korean Medicine (KM) Application Center, Korea Institute of Oriental Medicine (KIOM), Daegu 41062, Korea; jistr@kiom.re.kr (E.-J.L.); nhyim23@kiom.re.kr (N.-H.Y.); 2Research Group of Healthcare, Korea Food Research Institute, Wanju-gun 55365, Korea; truka@kfri.re.kr; 3Department of Food Biotechnology, Korea University of Science & Technology, Wanju-gun 55365, Korea

**Keywords:** small vessel occlusion, CYP26C1, DNA methylation, blood parameter, retinoic acid

## Abstract

The risk factors for stroke, a fatal disease, include type two diabetes, hypertension, and genetic influences. Small vessel occlusion (SVO) can be affected by epigenetic alterations, but an association between SVO and the methylation of cytochrome P450 family 26 subfamily C member 1 (CYP26C1) has not been identified. In this study, we measured the level of DNA methylation in the CYP26C1 promoter and the 5′ untranslated region of 115 normal subjects and 56 patients with SVO in Korea. The DNA methylation level of each subject was measured by bisulfite amplicon sequencing, and statistical analysis was performed using the general linear model or Pearson’s correlation. The average level of DNA methylation was markedly lower in patients with SVO than in normal subjects (20.4% vs. 17.5%). We found that the methylation of CYP26C1 has a significant positive correlation with blood parameters including white blood cells, hematocrit, lactate dehydrogenase, and Na+ in subjects with SVO. We predicted that binding of RXR-α and RAR-β might be affected by CYP26C1 methylation at CpG sites −246–237 and −294–285. These findings suggest that CYP26C1 methylation in the promoter region may be a predictor of SVO.

## 1. Introduction

Stroke is a major cause of death in elderly subjects: the death rate from stroke in Korea is 48.5 per 100,000 persons [1]. In 2016, there were 13.7 million new patients with stroke worldwide [2]. Stroke phenotypes can be broadly divided into ischemia and hemorrhage, and nearly 80% of stroke is ischemic [2]. Ischemic stroke is subdivided into five subtypes, namely, large artery atherosclerosis (LAA), cardioembolism (CE), small vessel occlusion (SVO), stroke of other determined etiology, and stroke of undetermined etiology, according to the Trial of ORG 10,172 in Acute Stroke Treatment (TOAST) classification [3]. The incidence of SVO among ischemic stroke subtypes is higher than that of other subtypes in Asian individuals [4]. The mortality rate due to SVO is low compared with other stroke types [5,6], but SVO is known to increase the incidence of stroke recurrence, vascular dementia, and Alzheimer’s disease [7,8,9]. Therefore, ongoing care for SVO needs to include the prevention of the occurrence of serious diseases.

The risk factors for stroke include type two diabetes, hypertension, genetic factors, and environmental factors such as smoking [10,11,12]. Recently, many studies have been conducted into the correlation between epigenetic changes, which are affected by environmental factors, and stroke. These studies aim to identify the molecular mechanisms underlying the effects of epigenetic variation on the occurrence of stroke [13,14,15].

Cytochrome P450 family 26 subfamily C member 1 (CYP26C1) is a member of the cytochrome P450 family and is involved in the catabolism of retinoic acid (RA) [16]. Until now, the relation between CYP26C1 and stroke has not been understood, but some important studies into RA and stroke suggest that the function of CYP26C1 might be affected by stroke, leading to poor outcomes [17,18,19,20]. Cai et al. demonstrated that the administration of all-trans RA before transient middle cerebral artery occlusion protects against cerebral ischemia in a mouse model [17]. Yu et al. showed that treatment with cis RA increases the recovery of motor function in rats with induced distal middle cerebral artery occlusion [18]. Another study by Yang et al. showed that all-trans RA is decreased by approximately 39% in patients with idiopathic dilated cardiomyopathy [19]. Because CYP26C1 catalyzes the degradation of all-trans RA, these studies suggest the hypothesis that variation in CYP26C1 activity and epigenetic chromosomal changes in CYP26C1, which affect the mRNA expression of CYP26C1, might be associated with stroke.

In the current study, we investigated the DNA methylation level in normal individuals and patients with SVO and investigated the association between epigenetic alterations in the CYP26C1 promoter region and SVO in Korean subjects.

## 2. Materials and Methods

### 2.1. Subjects

Fifty-five patients with acute stroke who had experienced a first stroke within one week, and were diagnosed as having SVO, were enrolled in this study. The stroke type was determined by either computed tomography or magnetic resonance imaging. One hundred fifteen normal subjects without SVO, or any other symptoms related to SVO, were also enrolled. Subjects with histories of hypertension, diabetes, hyperlipidemia, coronary heart disease, or stroke were excluded from the study.

This study was conducted as the follow-up study to “The Fundamental Study for the Standardization and objectification of PI in TKM for stroke”—a project by the Korean Institute of Oriental Medicine (KIOM) [21]—and was approved by the Institutional Review Board (IRB No. I-1603/001-001-01) of the KIOM.

After obtaining written informed consent, blood for DNA extraction and for analysis of blood parameters were collected from each of the subjects, and blood parameters were analyzed using an automatic biochemistry analyzer.

### 2.2. DNA Methylation Analysis

Genomic DNA from the blood of each subject was extracted using GeneAll Genomic DNA Extraction Kits (GeneAll, Seoul, Korea). DNA methylation was analyzed using BSAS, performed by the Life is Art of Science Laboratory (Gimpo, Korea). Briefly, DNA was bisulfite-converted using EZ DNA Methylation Kits (Zymo Research, Irvine, CA, USA) according to the manufacturer’s protocol, and the CYP26C1 promoter region at 94,820,155–94,820,614 (−400 bp–+60 bp from transcriptional start site) was amplified using specific PCR primers as follows: gene-F, 5′-GAGAGAAAGGTATTTAGAAGTT-3′ and gene-R, 5′-ATTACCAACCTAATCTACAAACCCC-3′. After constructing a PCR library of PCR products using Illumina TruSeq Nano DNA Sample Prep Kits (Illumina, San Diego, CA, USA), the library was sequenced on a MiSeq system (Illumina). Potentially existing adapter sequences and raw qualities based on the low read were trimmed using Skewer Ver 0.2.2 [22]. Mapping and calling of the methylation levels was performed using BS-Seeker2 software, according to the UCSC hg19 reference sequence [22].

### 2.3. Prediction of Transcription Factors in CpG Sites

To identify transcription factors which can bind to the CpG sites located in the CYP26C1 promoter region, we searched for putative TFBS using the PROMO database [23]. The prediction of transcription factors was conducted only for humans and was limited to 5% dissimilarity.

### 2.4. Statistical Analysis

Statistical analysis was performed using SPSS 19.0 (Seoul, Korea). Chi-squared tests were used to compare the differences between categorical variables. The normality of continuous variables was tested using Kolmogorov–Smirnov tests, and the differences in continuous variables were compared using Student’s *t*-tests or a binary general linear model adjusted for sex, age, smoking, drinking, body mass index (BMI), and waist-hip ratio (WHR). Two-way Pearson correlations were performed to measure the correlations between the DNA methylation and the blood parameters. Statistical significance was determined as *p* < 0.05.

### 2.5. Availability of Data and Materials

All datasets used and analyzed in the current study are available from the corresponding author upon reasonable request.

### 2.6. Consent to Participate and Ethics Approval

Subjects who participated in this study provided written informed consent. This study was approved by the Institutional Review Boards (IRB No. I-1603/001-001-01) of the KIOM.

## 3. Results

### 3.1. Characteristics of Subjects

The clinical characteristics of the subjects are shown in Table 1. A total of 171 subjects (115 normal subjects and 56 patients with SVO) participated in this study. The waist–hip ratio was higher in patients with SVO than in normal subjects (*p* < 0.001). In addition, white blood cells (WBCs), platelets, glutamic oxaloacetic transaminase (GOT), glutamic pyruvate transaminase (GPT), triglyceride, and fasting blood sugar (FBS) were significantly higher in patients with SVO than in normal subjects (WBC, *p* = 0.002; platelet, *p* = 0.002; GOT, *p* = 0.105; GPT, *p* = 0.034; triglyceride, *p* = 0.907; FBS, *p* < 0.001). In contrast, both total cholesterol and high-density lipoprotein cholesterol were lower in patients with SVO than in normal subjects (*p* = 0.106 and *p* = 0.002, respectively).

### 3.2. Map of CpG Islands in the CYP26C1 Promoter Region

We confirmed the methylation of CpG islands in CYP26C1 using a genomic map of CYP26C1 (NM_183374.3). CYP26C1 is located on the positive strand of chromosome 10, and CpG island (CGI:255) exists in the promoter region. Forty-three CpG sites were identified from a fragment in the 5′-upstream region of the CYP26C1 promoter at 94,820,155–94,820,614 (Figure 1).

### 3.3. CYP26C1 Promoter Methylation in Normal Subjects and in Patients with SVO

We found hypomethylation of 43 CpG islands in patients with SVO and normal subjects through pilot studies using whole bisulfite amplicon sequencing (BSAS) analysis (Appendix A). The mean of DNA methylation levels at these 43 CpG sites was 0.175 ± 0.046 in patients with SVO, which is significantly lower than the mean 0.204 ± 0.039 in normal subjects (*p* < 0.0001) (Figure 2A). DNA methylation levels were significantly lower in patients with SVO at CpG2 (*p* = 0.0136), CpG11 (*p* = 0.0006), CpG16 (*p* = 0.0134), CpG19 (*p* = 0.0002), CpG25 (*p* = 0.0002), and CpG35 (*p* = 0.0013) than in normal subjects (Figure 2B). The results suggested that CYP26C1 promoter methylation is associated with SVO.

### 3.4. Association of CYP26C1 Promoter Methylation with Blood Parameters

We further investigated the association between the DNA methylation of the CYP26C1 promoter region and the blood parameters of the subjects. WBC and Hct among the blood parameters including WBC (*r* = −0.270, *p* = 0.004), hematocrit (Hct; *r* = −0.193, *p* = 0.039), and lactate dehydrogenase (LDH; *r* = −0.221, *p* = 0.019) in normal subjects. Na^+^ (*r* = 0.191, *p* = 0.041) showed significant positive correlation with DNA methylation in normal subjects (Figure 3A). WBC (*r* = −0.337, *p* = 0.004) and LDH (*r* = −0.334, *p* = 0.040) also showed negative correlation in SVO patients (Figure 3B). Correlation between each CpG site and blood parameters is presented in Appendix A Appendix A. These findings suggest that the methylation of CYP26C1 has a significant association with serum WBC and Hct and LDH in subjects.

### 3.5. Binding Sites of Transcription Factors at CpG Sites in the CYP26C1 Promoter Region

Changes to the methylation status of DNA fragments at CpG sites in the CYP26C1 promoter region can affect the binding affinity of transcription factors for targeted DNA sequence, which can consequently affect gene expression. Therefore, we analyzed the transcription factor binding sites (TFBS) using PROMO to identify specific transcription factors that bind to the CpG sites of CYP26C1, which may therefore be affected by methylation. As shown in Figure 4, the presence of binding sites in the CYP26C1 methylation promoter region was confirmed in various CpG sites. Of these sites, binding of RXR-α and RAR-β might be affected by methylation at CpG sites of −246–237 and −294–285 position. All predictive transcription factors were listed in Appendix A.

## 4. Discussion

Stroke is one of the major causes of death, mortality, and disability worldwide and incurs high costs for treatment and post-stroke care. It is a multifactorial disease, and genetic and epigenetic characteristics could play a role in the occurrence and development of stroke. Epigenetic alterations—chromosomal modifications without changes to the DNA sequence—play important roles as physiological mediators of development and cellular functioning in ischemic stroke [24]. Some research into epigenome-wide methylation has used hypothesis-free analysis to evaluate the level of DNA methylation at specific sites throughout the genome, with the aim of discovering ischemic stroke-associated methylation sites, without bias toward specific loci [14]. Several studies have reported that specific DNA methylation levels are associated with ischemic stroke [25]. For example, Bushueva et al. assessed DNA hypomethylation in the *MPO* gene that is associated with cerebral stroke [26]. Another study by Li et al. identified that genes involved in the homocysteine metabolic pathway—methylenetetrahydrofolate dehydrogenase 1, cystathionine β-synthase, and dihydrofolate reductase—were hypomethylated in patients with stroke as compared with patients with hypertension [27]. Homocysteine is an important intermediate in DNA methylation through the methionine–homocysteine cycle [28], and DNA methylation changes of genes involving in the homocysteine metabolic pathway might affect homocysteine level. A high level of homocysteine is known to increase SVO [29].

In the present study, we investigated the methylation of the CYP26C1 promoter region in patients with SVO and in normal subjects and investigated the association between the methylation status and SVO. CYP26C1, located at chromosome 10 q23.33, has one CpG island, CpG:255, spanning the promoter and exon 1 region (Figure 1A), and its CpG island includes 41 CpG sites in the region upstream of the transcription start site (Figure 1B). The mean DNA methylation level in the SVO group was 0.175, which was significantly lower than the level in the normal group, 0.204 (Figure 2), and almost all CpG sites were hypomethylated in the SVO group (Appendix A). Among the blood parameters, methylation was negatively correlated with WBC, Hct, and LDH, but positively correlated with blood Na^+^ level (Figure 3).

CYP enzymes act by regulating the expression levels of genes involved in the synthesis and degradation of virtually all nonprotein ligands that bind to receptors or activate the secondary messenger pathways that regulate growth, differentiation, apoptosis, homeostasis, and neuroendocrine function [30]. Of these, three members of the *CYP26* gene family, *CYP26A1*, *CYP26B1*, and *CYP26C1*, are involved in the degradation of RA to hydroxy-RA [31].

Until now, the effect of CYP26C1 on stroke has not been fully understood, but several studies into the effects of RA in ischemic stroke have provided indirect evidence that CYP26C1 might be associated with ischemic stroke. Cai et al. showed that the all-trans RA level in neutrophils in mice with stroke was significantly decreased compared with a control group [17]. A human study reported by Yu et al. showed that higher plasma RA levels delay the occurrence of the first ischemic stroke in a hypertensive Chinese population [32]. Another study performed by Guo et al. also showed that 9-cis-retinal, a metabolite of the RA metabolism pathway, is significantly lower in patients with stroke than in normal individuals [33]. RA also lowers the plasma level of homocysteine, intermediate in DNA methylation, in rats [34]. These studies suggested that a change of CYP26C1 expression by DNA methylation in the promoter region might affect ischemic stroke by regulating the RA level.

In the current study, we identified CpG:255 in the CYP26C1 promoter region as being related to SVO, possibly via RA degradation. Generally, epigenetic variations such as DNA methylation in the promoter region affect the binding activity of transcription factors, which regulate the expression of the genes [35,36]. Based on the PROMO database, we selected data about predicted transcription factors that could bind to CpG sites in the CYP26C1 promoter region, showing that this CpG island has binding sites for the retinoic acid receptors (RAR) retinoid X receptor (RXR) dimers, CREB, and c-JUN (Figure 4). RA binds to the cellular RA-binding protein and can be transferred to the nuclear RAR [37]. RAR regulates the transcription of genes by heterodimerization with the RXR [38]. It has also been reported that RA upregulates the transcription factors of c-JUN by increasing ERK1/2 activity, which increases protein kinase A activity to induce the phosphorylation of CREB [39]. According to these studies, the expression of CYP26C1 might be regulated by RA-linked transcription factors. Taimi et al. showed that RA induces CYP26C1 expression in a HPK1a cell line, and Zolfaghari et al. reported that RAR-mediated signaling induces the expression of CYP26A1 (one of the CYP26 family) [16,40]. These data suggest that hypomethylation of CYP26C1 might induce its expression and might decrease regional RA, which affects the occurrence of SVO.

The limitation of this study is, firstly, the small sample size. Because the subjects enrolled in this study were elderly, and the number of patients with stroke was small, it is difficult to generalize the association between CYP26C1 and ischemic stroke. Secondly, the patients with stroke enrolled in the study were only patients with SVO. Studies on the association between other types of SVO, such as LAA and CE, are needed. Thirdly, follow-up studies are needed to identify the effect of methylation on CYP26C1 expression and RA level. Despite these limitations, to the best of our knowledge, this is the first study to explore the association between promoter methylation of CYP26C1 and ischemic stroke. Further studies with a large and diverse population should be performed to confirm and generalize our results.

## 5. Conclusions

In conclusion, we evaluated the DNA methylation of the CYP26C1 promoter region. We found an association between hypomethylation in this region and SVO in elderly Korean subjects. We also identified an association between methylation at specific CpG sites in the promoter region of CYP26C1 and blood parameters related to SVO. We further showed that RXR-α and RAR-β might be affected by CYP26C1 methylation. From these findings, we conclude that CYP26C1 methylation may be an epigenetic potential indicator of SVO.

## Figures and Tables

**Figure 1 genes-12-01622-f001:**
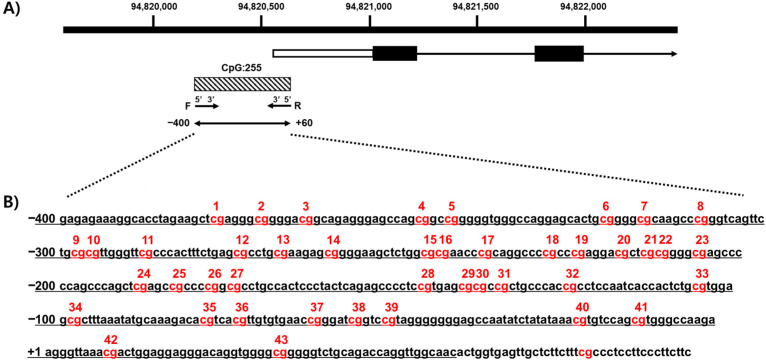
Analysis of methylation of CpG sites in the CYP26C1 promoter region. (**A**) The black box and arrow show CYP26C1, and the shaded box indicates the CpG island region located in the 5′-flanking region of CYP26C1. (**B**) The sequence of the CYP26C1 promoter region. The underline indicates the positions of DNA methylation. The CpG sites analyzed in this study are numbered. Red: DNA methylation of CpG sites.

**Figure 2 genes-12-01622-f002:**
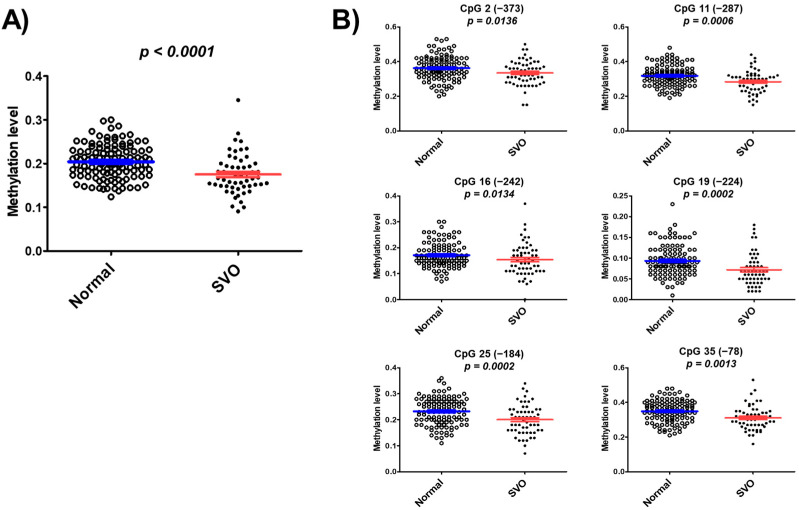
Methylation of CYP26C1 is associated with SVO. (**A**) Distribution of DNA methylation levels in normal subjects and patients with SVO. (**B**) Significance of hypomethylation at the specific CpG sites of the CYP26C1 promoter region in patients with SVO, compared with normal subjects. Red line and Blue line: Mean ± standard error.

**Figure 3 genes-12-01622-f003:**
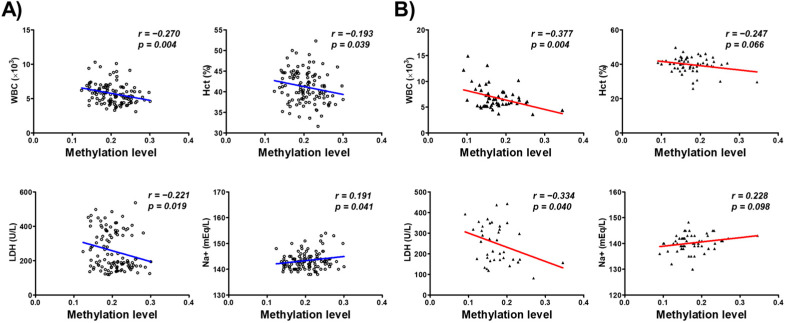
Analysis of the association between methylation of the CYP26C1 promoter region and blood parameters in normal and SVO subjects. WBC—white blood cell; Hct—hematocrit; LDH—lactate dehydrogenase; Na^+^—natrium. (**A**): Normal subjects; (**B**): SVO subjects. Red line and Blue line: linear regression between methylation of the CYP26C1 promoter region and blood parameters.

**Figure 4 genes-12-01622-f004:**
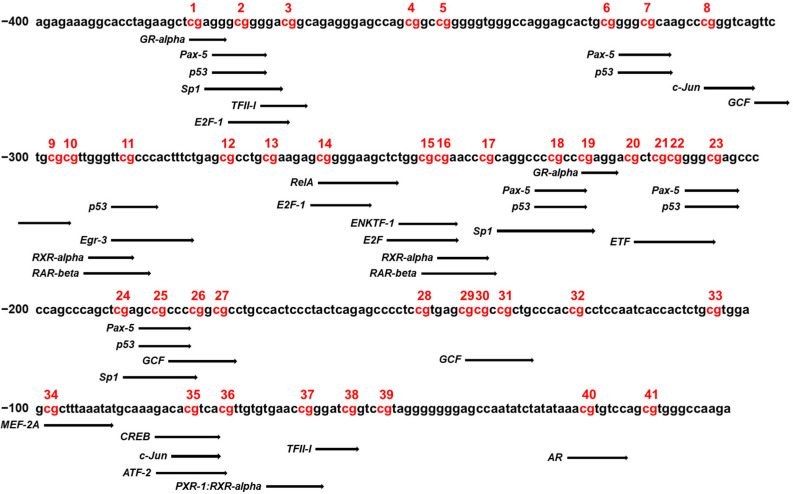
Analysis of the transcription factor binding sites in the CYP26C1 promoter region. PROMO/TFBS analysis of the sequences of CpG sites in the CYP26C1 promoter region. Each black arrow indicates a specific transcription factor that matches the predicted binding sequence. Red: DNA methylation of CpG sites.

**Table 1 genes-12-01622-t001:** General characteristics of normal subjects and of patients with small vessel occlusion (SVO).

Characteristics	Normal	SVO	*p*-Value
*n*	115	56	
*Anthropometric characteristics*			
Sex (M/F)	59/56 ^a^	37/19	0.048 *
Age (years)	60.41 ± 11.06 ^b^	61.74 ± 12.99	0.489 ^#^
Smoking (none/stop/active)	68/29/18	23/7/26	<0.001 *
Drinking (none/stop/active)	59/9/47	24/5/26	0.645 *
Weight (kg)	58.94 ± 7.64	58.81 ± 8.15	0.917 ^#^
BMI (kg/m^2^)	22.59 ± 1.39	22.31 ± 1.38	0.225 ^#^
Waist circumference (cm)	80.92 ± 6.89	80.18 ± 5.29	0.526 ^#^
WHR	0.872 ± 0.055	0.914 ± 0.061	<0.001 ^#^
*Medical history*			
Depression (yes, %)	1 (0.9) ^a^	1 (1.8)	0.549 *
Migraine (yes, %)	12 (10.4)	7 (12.5)	0.434 *
*Blood parameters*			
WBC (×10^3^)	5.7 ± 1.46	6.85 ± 2.21	0.003 ^&^
RBC (×10^6^)	4.47 ± 0.43	4.36 ± 0.52	0.021 ^&^
Hg (g/dL)	13.79 ± 1.33	13.37 ± 1.62	0.001 ^&^
Hct (%)	41.17 ± 3.80	39.67 ± 4.58	0.001 ^&^
Platelet (×10^3^/μL)	196.92 ± 68.527	230.05 ± 61.21	0.002 ^&^
GOT (U/dL)	23.03 ± 6.88	25.68 ± 12.58	0.053 ^&^
GPT (U/dL)	19.26 ± 8.26	23.40 ± 19.06	0.008 ^&^
Total cholesterol (mg/dL)	200.30 ± 38.60	188.64 ± 43.34	0.147 ^&^
Triglyceride (mg/dL)	127.18 ± 64.56	132.81 ± 55.44	0.896 ^&^
HDL cholesterol (mg/dL)	56.97 ± 13.15	48.50 ± 14.41	0.005 ^&^
LDL cholesterol (mg/dL)	119.09 ± 32.98	115.88 ± 38.51	0.0.869 ^&^
FBS (mg/dL)	96.89 ± 9.02	113.707 ± 33.14	<0.001 ^&^
LDH (U/L)	255.68 ± 111.30	246.94 ± 95.71	0.230 ^&^
Na^+^ (mEq/L)	143.41 ± 3.26	140.23 ± 3.40	<0.001 ^&^
K^+^ (mEq/L)	4.39 ± 0.37	4.16 ± 0.63	0.004 ^&^

^a^ Indicates the number of subjects (%). ^b^ Indicates the mean ± standard deviation. Categorical variables were analyzed using the chi-squared tests or Fisher’s exact tests *, and continuous variables were analyzed using Student’s *t*-tests ^#^ or a binary general linear model ^&^, after adjusting for sex, age, smoking, drinking, BMI, and WHR. BMI—body mass index; WHR—waist–hip ratio; WBC—white blood cell; RBC—red blood cell; Hg—hemoglobin; Hct—hematocrit; GOT—glutamic oxaloacetic transaminase; GPT—glutamic pyruvate transaminase; HDL cholesterol—high-density lipoprotein cholesterol; LDL cholesterol—low-density lipoprotein cholesterol; FBS = fasting blood sugar; LDH = lactate dehydrogenase.

## Data Availability

Not applicable.

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
