# Peer review of "Association of CYP26C1 Promoter Hypomethylation with Small Vessel Occlusion in Korean Subjects"

_genes, 2021, doi:10.3390/genes12101622_

Round 1

Reviewer 1 Report

It is well known, that vascular events, such as stroke or heart attacks, are accompagnied by an increase homocysteine levels. This elevation reflects an impairment of the methylation potential and capacity. Therefore DNA hypomethylation occurs. The same occurs also in levodopa- or valproic acid treated patients. Therefore i recommend that authors add this remark in the concluding remarks and discuss their outcomes as epiphenomenon. 

see also: 

Movement Disorders DOI: 10.1002/mds.26560

Role of homocysteine in the treatment of Parkinson's disease. Expert Rev Neurother. 2008 Jun;8(6):957-67. doi: 10.1586/14737175.8.6.957. PMID: 18505360.

Reviewer 2 Report

This article by EJ Lee et al. is the first publication that shows a co-relation between the methylation levels of CYP26C1 and Small Vessel Occlusion (SVO), a subtype of ischemic stroke which risk factors are poorly characterized. I found really interesting the approach made by the authors focus on searching the Transcription Factors associated to the promotor region studied that can be affected by methylation. This is an approach really interesting at molecular level that it is not been done previously. Thus, the finding presented in this manuscript are a great contribution to understanding the role of epigenetics on ischemic stroke. However, despite its novelty, this article needs to solve some issues.

  1. In the manuscript authors refers to their results to be associated with ischemic stroke meanwhile this study is done only in SVO patients. This should be changed because, even if the results are significant for the SVO, they cannot be generalized for all ischemic strokes.
  2. In the manuscript it specified that this study includes 56 ischemic stroke patients and 115 healthy subjects (line 17, 111-112). However, on line 64 it is specified 55 patients and on Table1 the number of healthy (normal) individuals is 104 and 80 for ischemic stroke patients. Please clarify the exact number of patients included in the study and, if there is been exclusion of some patients, specify why in the manuscript.
  3. It is not clear in the manuscript when the blood samples were extracted. When performing a methylation study, a huge variability on the time of obtaining the samples can affect the results as the methylation status can change over time. Please specify this on the methods section. If the time of extraction is different among samples, it should be specified and showed on Table 1. This is also important in order to replicate the results in future studies.
  4. Additionally, it should be specified in the method section if the data of the Blood parameters correspond to the time of extraction or is from the admission of the stroke patients to the emergency room.
  5. Line 81: The size of the promoter region amplified could be informative an important to replicate the results by other groups, I consider that should be included in the manuscript. Additionally, it would be informative to include the location of the primers used in Figure 1A.
  6. I consider that Figure 1B will be more reader-friendly if the number of at least the significant CpG were included in the figure.
  7. Line 98-99 and line 123: The general lineal model should include all the significant variables associated with ischemic stroke in their cohort sample in addition to the standard variables of age and sex in order to show a real independent association
  8. On Table 1 the majority of p-values do not show any ‘a, b, c’ letter that indicates the type of statistical analysis done. Should the reader assume that the numbers below the first one that shows a letter had been done using the same approach? Please clarify this on table legend or add the corresponding letters to each p-value.
  9. Line 143-144: This is the main comparison of this study. I consider that it should be indicated the exact p-value in the manuscript. Additionally, I think the individual CpG p-value should be adjusted by Bonferroni because there are several CpG sites included in the analysis.
  10. Line 156-159: It is not clear that those correlations and p-values correspond to the association to ischemic stroke patients I will recommend to rephrase that. Additionally, if the methylation is truly correlated with blood parameters they should be able to use also the normal subjects included. I consider that the correlation in normal patients could be shown in Figure 3 together with ischemic stroke data using different color or dot shape. I understand that this analysis is done with the average DNA methylation, but I think it could be interesting to see or mention the individual significant CpG sites (included in Table S2).

Furthermore, the authors look for a correlation in all the blood parameters with the methylation, but the aim of this approach is not well explain in the manuscript. I will also recommend to focus the correlation analysis only in the Blood parameters that show differences between stroke patients and controls.

  1. Line 168: there is a full stop ‘.’ missing between ‘gene expression’ and ‘Therefore’.
  2. Line 173: The authors refer to the CpG positions indicating its location, but it will be more reader-friendly if they refer to them as numbered CpGs, like they had done previously on the manuscript (line 145-147). In this way is more easily to relate the results of the DNA methylation analysis with the finding of Transcription Factor binding sites. Additionally, as in Figure 1B, Figure 4 should include labels indicating the CpG number and/or a good scale ruler to locate better the CpG sites referred in the manuscript.
  3. Line 218: I do not agree with this statement. This study provides some useful information to relate CYP26 methylation with stroke, but it is not clear if this methylation site can down regulate the expression of CYP26. In order to declare that they should show correlation of this CpG island with CYP26C1 mRNA levels or eQTLs.
  4. Line 252: The authors statement that they had verified that RXR-alpha and RAR-beta might be affected by CYP26C1 methylation. However, I do not agree with that. They result suggest a possible rol of RXR-alpha and RAR-beta in the regulation of CYP26C1 during Small Vessel Occlusion. However, to verify this further studies are needed, such as target mutagenesis of the TFs binding sites together with ChIP or gel swift assays.
  5. Line 253-254: In order to consider the methylation levels of the CYP16C1 promoter a biomarker, the authors need to show an independent association of the DNA methylation levels to ischemic stroke. And, to further confirm this, it is necessary to stablish a cut-off value that will predict the incidence of ischemic stroke. Furthermore, this study is done only in SVO patients, so if they do the independent association it will be a risk factor of SVO, but it cannot be generalized for all ischemic stroke subtypes.
  6. Table S2 and S3 are not referred in the manuscript. I think the information provided in those Tables is informative and should be referenced in the manuscript.

Round 2

Reviewer 1 Report

Accept

Reviewer 2 Report

The authors addressed all the issues reported. I do not have any further comments.